# ACTIVELY LEARNING HORN ENVELOPES FROM LLMS

## ABSTRACT

We propose a new strategy for extracting Horn rules from Large Language Models (LLMs). Our approach is based on Angluin's classical exact learning framework where a learner actively learns a target formula in Horn logic by posing *membership* and *equivalence* queries. While membership queries naturally fit into the question-answering setup of LLMs, equivalence queries are more challenging. Previous works have simulated equivalence queries in ways that often lead to a large number of uninformative queries, making such simulations prohibitively expensive for modern LLMs. Here, we propose a new adaptive prompting strategy for posing equivalence queries, making use of the generative capabilities of modern LLMs to directly elicit counterexamples. We consider a case study where we learn rules describing gender-occupation relationships from the LLM. We evaluate our approach on the number of queries asked, the alignment of the extracted rules with historical data, and the consistency across runs. We show that our approach is able to extract Horn rules aligned with historical data with high confidence while requiring orders of magnitude fewer queries than previous methods.

## 1 INTRODUCTION

Large language models (LLMs) have in recent years displayed remarkable capabilities when interacting with users and answering queries. However, as LLMs are trained on large amounts of data, the information that is retained after training, the knowledge embedded in the model, becomes obscured. An important area of research in recent years has therefore been focused on assessing or extracting the knowledge acquired by LLMs (Roberts et al., 2020; Petroni et al., 2019; He et al., 2025). One can see the task of extracting knowledge from an LLM as an *active learning* task, where the learner interacts with a teacher by posing queries. The basic idea is that, by asking the right queries based on the answers received, the learner can navigate the space of possibilities and obtain the desired knowledge from the teacher more efficiently.

The exact learning framework by Angluin (1987) formalizes this mode of learning in a mathematical way. The most studied communication protocol between the learner and the teacher, also called an *oracle*, consists of *membership* and *equivalence queries*. In set theory notation, a membership query asks 'Is $x \in \mathcal{T}$?' where $\mathcal{T}$, known as the *target*, is a set that represents what the learner wants to learn. In an *equivalence query*, the learner sends its idea of what the target is, the *hypothesis* $\mathcal{H}$, and asks if $\mathcal{H}$ is equivalent to $\mathcal{T}$. In set theory notation, 'Is $\mathcal{H} = \mathcal{T}$?'. The answer is 'yes' if they are equivalent or 'no' together with a *counterexample* illustrating the difference, that is, a counterexample $x \in \mathcal{H} \oplus \mathcal{T}$, where $\oplus$ is the symmetric difference. We say that the counterexample is *negative* if it models the hypothesis but not the target, and *positive* if it models the target but not the hypothesis.

In this work, we implement the task of extracting rules from LLMs as a learning task within Angluin's exact learning framework. The membership query asks whether a statement is possible in the "view" of the LLM, to which the LLM should answer 'yes' or 'no' (e.g., 'is it possible for astronauts to be born in Antarctica?'), and we formalize these expressions using propositional Horn logic (e.g. $\neg(\mathsf{astronaut} \wedge \mathsf{Antarctica})$). We collect the answers of the LLM to form a *hypothesis*, a Horn formula that expresses the knowledge extracted so far. Membership queries naturally fit into the question-answering setup of LLMs but answering equivalence queries is more complex in this setting. The equivalence query requires the teacher to understand the given hypothesis, compare with the target representing its "view" of the world, and if it is not equivalent, return a counterexample illustrating where the hypothesis and the target differ.

Blum et al. (2024) extracted Horn expressions from BERT-based models (Devlin et al., 2018; Liu et al., 2019) using random sampling to simulate equivalence queries. The idea behind simulating equivalence queries this way is to ask the model to classify randomly generated examples as positive ('yes') or negative ('no') with membership queries. If the learner finds an example $x$ that is classified differently by the model than expected from our hypothesis, then the learner can proceed as if the teacher had replied 'no' to an equivalence query and returned $x$ as counterexample. However, this simulation strategy has several drawbacks, as random generation may create several uninformative queries before finding a counterexample. We avoid simulating equivalence queries with random sampling by *directly* querying for a counterexample to a given hypothesis. Considering the generative capability of LLMs, we explore their ability to parse the hypothesis and provide a valid counterexample if the hypothesis differs from the target. Although parsing logical information from LLM responses has been previously explored (Ye et al., 2023), our work is, to the best of our knowledge, the first to explore this with the purpose of generating counterexamples to equivalence queries. Our algorithmic strategy is based on the algorithm by Blum et al. (2024), which is a non-trivial adaptation of the algorithm for exact learning Horn expressions by Angluin et al. (1992). The adapted algorithm returns a *Horn envelope*, a Horn expression that is the closest approximation of the true target. To evaluate our approach, we consider a case study on gender-occupation relationships in LLMs.

The main contributions of this work are as follows: (i) an implementation of the algorithm for learning Horn envelopes Blum et al. (2024) adapted to learn from generative LLMs; (ii) an adaptive prompt generation strategy, based on the current working hypothesis built using the responses of the LLM, that implements, not simulates, equivalence queries by directly asking the LLM to generate counterexamples; (iii) an experimental analysis of gender-occupation relationship rules extracted from LLMs, where we assess the impact of model type and model size of the rules extracted and analyze the consistency and confidence of extracted rules.

In the following section, we describe related works on using machine learning (ML) models as teachers and as reasoners. In Section 3, we provide basic notions and notation used in this paper. In Section 4, we discuss some challenges in extracting rules (in the format of Horn expressions) from LLMs, in particular, we describe our prompting strategy for dealing with equivalence queries. We describe our experiments in Section 5 and the evaluation criteria and results in Section 6. Finally, we conclude in Section 7.

## 2 RELATED WORKS

**ML models as teachers** Previous works have explored the idea of using a machine learning model as a teacher within Angluin's exact learning framework to extract information from the model. Weiss et al. (2024) employed the exact learning framework to extract automata from recurrent neural networks (RNNs). They used an algorithm designed to pose membership and equivalence queries. Since RNNs cannot naturally answer equivalence queries, these queries were simulated in two ways: by random sampling and by using a heuristic to find counterexamples. As previously mentioned, Blum et al. (2024) extracted Horn expressions from BERT-based models and it is this work that comes closest to ours. They conducted a case study on gender bias in occupations and extracted Horn expressions manifesting biases in the BERT-based models. The same study has also been taken as basis in the work by Ozaki et al. (2025), which extracts decision trees, also indicating the presence of occupational-based gender biases in these models.

**ML models as reasoners** There have been many survey papers on reasoning in LLMs (Mondorf & Plank, 2024; Huang & Chang, 2023; Sun et al., 2024) which explore reasoning behaviours in LLMs and highlights the interest in the field. Previous works have employed many tools in aiding LLMs in reasoning tasks. As mentioned Ye et al. (2023) used LLMs to parse logical information from the query but they leveraged an outside theorem prover to derive the final answer. Our work uses an algorithm to derive our hypothesis but still relies on the model to parse our hypothesis and use reasoning to provide a counterexample. Brown et al. (2020) explored how few-shot prompting could aid reasoning in larger models and this is the method used in our work.

## 3 PRELIMINARIES

We introduce the notation and basic relevant notions about propositional logic, Horn envelopes, and the exact learning framework.

**Propositional Logic, Horn Envelopes** Let $\mathcal{P}$ be a set of propositional symbols. A *literal* is an element $p$ of $\mathcal{P}$ or its negation, denoted $\neg p$. A literal is *positive* if it is in $\mathcal{P}$, otherwise it is *negative*. A *clause* is a disjunction of literals, in symbols, $l_1 \vee \ldots \vee l_n$, with $n \in \mathbb{N}$ and each $l_i$ a (positive or negative) literal. A *Horn clause* is a clause with at most one positive literal. A *propositional expression* (in normal form) is a set of clauses. A *Horn expression* is a propositional expression that has only Horn clauses. The semantics of propositional expressions is given by *interpretations*. An interpretation is a subset of $\mathcal{P}$. An interpretation $\mathcal{I}$ *satisfies*: a positive literal $p \in \mathcal{P}$ iff $p \in \mathcal{I}$; a negative literal $\neg p$ iff $p \notin \mathcal{I}$; a clause iff $\mathcal{I}$ satisfies at least one of its literals; and a propositional expression iff it satisfies all of its clauses. Given a propositional expression $\phi$, we write $\mathcal{I} \models \phi$ if $\mathcal{I}$ satisfies $\phi$. We define interpretations$(\phi)$ as $\{\mathcal{I} \mid \mathcal{I} \subseteq \mathcal{P}, \mathcal{I} \models \phi\}$. Given propositional expressions $\phi, \psi$, we say that $\phi$ *entails* $\psi$, written $\phi \models \psi$, iff interpretations$(\phi) \subseteq$ interpretations$(\psi)$. Also, $\phi, \psi$ are *logically equivalent*, written $\phi \equiv \psi$, iff interpretations$(\phi) =$ interpretations$(\psi)$. The *closure under intersection* of a set of interpretations $M$ is the set of all interpretations that can be obtained as the intersection of interpretations in $M$. Let Pow$(S)$ denote the power set of a set $S$.

**Proposition 3.1 ((Dechter & Pearl, 1992))** *A set of interpretations $M \subseteq$ Pow$(\mathcal{P})$ is closed under intersection iff $M =$ interpretations$(\phi)$ for some Horn expression $\phi$.*

By Proposition 3.1, for every set of interpretations closed under intersections, there is a unique (up to logical equivalence) Horn expression that represents it. Given an arbitrary propositional expression $\phi$, the *Horn envelope* of $\phi$ is the Horn expression that represents the closure under intersection of the models of $\phi$. The Horn envelope of $\phi$, denoted env$(\phi)$, can be seen as the closest approximation of $\phi$ to a Horn expression. Syntactically, this Horn expression is assumed to be the Duquenne-Guigues basis (Guigues & Duquenne, 1986), which has the minimum number of rules. We often write clauses in the format of a *rule* $P \to Q$, where $P$ is the conjunction of negative literals (or $\top$ if none is negative) and $Q$ is the disjunction of positive literals (or $\bot$ if none is positive). E.g., $\neg p_1 \vee \neg p_2 \vee \cdots \vee \neg p_n$ becomes $(p_1 \wedge p_2 \wedge \cdots \wedge p_n) \to \bot$ and $\neg p_1 \vee \neg p_2 \vee \cdots \vee \neg p_n \vee q$ turns into $(p_1 \wedge p_2 \wedge \cdots \wedge p_n) \to q$. Intuitively, one can read $P \to Q$ as 'if all propositional symbols in $P$' hold then at least one of those in $Q$ needs to hold'. In case there are no positive literals, that is, $P \to \bot$, then we read it as 'a world where everything in $P$ holds is not possible'.

**Exact Learning** We briefly present the exact learning framework (Angluin, 1987), adapted to the case of learning the Horn envelope of a propositional expression from interpretations. Membership and equivalence queries in this case are defined as follows. Let $\mathcal{T}$ be a propositional expression. We call $\mathcal{T}$ the *target*, meaning that $\mathcal{T}$ represents the knowledge of the teacher. A *membership query for* $\mathcal{T}$ takes as input an interpretation $\mathcal{I}$ and returns 'yes' if $\mathcal{I}$ satisfies $\mathcal{T}$ and 'no' otherwise. In symbols, $\mathsf{MQ}_{\mathcal{T}}(\mathcal{I}) =$ yes if $\mathcal{I} \in$ interpretations$(\mathcal{T})$, otherwise $\mathsf{MQ}_{\mathcal{T}}(\mathcal{I}) =$ no. A *Horn equivalence query for* $\mathcal{T}$ takes as input a propositional expression $\mathcal{H}$—the hypothesis—and returns 'yes' if the Horn envelopes of $\mathcal{T}$ and $\mathcal{H}$ are equivalent, otherwise, it returns 'no' and a counterexample in the symmetric difference of the sets of interpretations satisfying $\mathcal{T}$ and $\mathcal{H}$. In symbols, $\mathsf{EQ}_{\mathcal{T}}^{\mathsf{Horn}}(\mathcal{H}) =$ yes if env$(\mathcal{T}) \equiv$ env$(\mathcal{H})$, otherwise, 'no' and an element $\mathcal{I} \in$ interpretations$(\mathcal{T}) \oplus$ interpretations$(\mathcal{H})$. If the target and the hypothesis are Horn expressions then the notion of a Horn equivalence query coincides with the notion of equivalence query by Angluin et al. (1992), however, we cannot guarantee or expect the target to be a Horn expression when treating a machine learning model as the teacher. An algorithm *exactly learns* Horn expressions (with membership and Horn equivalence queries) if it always terminates returning a hypothesis equivalent to the Horn envelope of the target expression.

## 4 ACTIVELY LEARNING FROM LLMS

Extracting rules from LLMs is a challenging task as LLMs can struggle with logical reasoning (Mondorf & Plank, 2024) and are sensitive to changes in the prompt (Chen et al., 2024). For us to actively learn from the LLMs we need the model to be able to parse our hypothesis, compare with the target hypothesis, and reason about counterexamples in the symmetric difference between the hypothesis and the target. We describe the strategies employed to address the following challenges.

1. Translation: the format of the membership and equivalence queries needs to be adapted from interpretations and logical formulas into expressions in natural language, requiring translation from logic to natural language.
2. Format: we may ask the LLM to reply in a certain format, but there is no guarantee that the LLM will reply in the requested format, requiring us to validate responses to ensure a correct translation between logic expressions and natural language.
3. Validation: even if the LLM replies in the correct format, the reply may be incorrect, inconsistent with previous responses, or even change if the same query is posed multiple times.

Each of these challenges are carefully considered in our work. We start by explaining how we address the first challenge in the paragraph on *membership queries*. We describe how we address the two challenges within the paragraph on *equivalence queries* and in the paragraph on *Validating EQ responses*. Only once all challenges have been addressed are we able to translate the response from the model back into a logical expression for the algorithm.

**Learning Horn Envelopes** Angluin et al. (1992) presented a classical algorithm that exactly learns any target Horn expression in polynomial time. However, the polynomial bound is under the assumption that the target is indeed a Horn expression. This can no longer be assumed when we take an LLM as a teacher, and the classical algorithm is not guaranteed to terminate without the assumption. The main theoretical contribution of the work by Blum et al. (2024) was to propose an algorithm, called Horn Envelope, that handles the case where the unknown target is not (equivalent to) a Horn expression. The algorithm can be seen in the Appendix as Algorithm 2.

**Membership Queries** The role of the membership query is to allow us to uncover the model's implicit knowledge and iteratively refine the rules that form the hypothesis of the learner. E.g., asking whether an interpretation that contains the symbols Antarctica, before_1900 satisfies the "view" of the LLM, represented by $\mathcal{T}$, would correspond to the query $\mathsf{MQ}_{\mathcal{T}}(\{\mathsf{Antarctica}, \mathsf{before\_1900}\})$, which would translate to the prompt: "Has 'A person born before 1900 in Antarctica' been possible in the real world? Reply with 'It is possible' if yes and 'it is NOT possible' if no. Do not specify which values are different"[1].

**Equivalence Queries** The role of the equivalence query is to check if our hypothesis is equivalent to the target, and, if not, receive counterexamples that iteratively brings us closer to the target. If our current hypothesis was that no one had been born in Antarctica before 1900, then the query of if this hypothesis was equivalent to the the LLM's "view" of the world, represented by $\mathcal{T}$, would correspond with $\mathsf{EQ}_{\mathcal{T}}^{\mathrm{Horn}}(\{\neg(\mathsf{Antarctica} \wedge \mathsf{before\_1900})\})$, which would translate to appending: "The updated hypothesis: 'A person born in any continent, in any time period, who is any occupation, could be any gender.' describes the real world accurately, except A person born in Antarctica before the year 1900 is not possible. Please provide another counterexample to my hypothesis if possible" to the base equivalence query prompt. The complete few-shot equivalence prompt can be seen in appendix A.2.

**Example** Let the current hypothesis be empty, meaning that anything is possible. If the model contains the information that there were no astronauts born before 1900 then it could provide us with this information as a counterexample represented by the interpretation $\mathcal{I} = \{\mathsf{astronaut}, \mathsf{before\_1900}\}$. Since this interpretation does not satisfy the target, we want to construct a hypothesis that is also not satisfied by this interpretation. The Horn Envelope algorithm does this by adding the Horn clause $\neg(\mathsf{astronaut} \wedge \mathsf{before\_1900})$ to its hypothesis. In the next iteration, if the model responds to $\mathsf{EQ}_{\mathcal{T}}^{\mathrm{Horn}}(\{\neg(\mathsf{astronaut} \wedge \mathsf{before\_1900})\})$ with the interpretation $\mathcal{J} = \{\mathsf{astronaut}, \mathsf{Antartica}\}$ as a counterexample, since counterexamples are from the symmetric difference of the interpretations satisfying the target and the hypothesis, we know that this counterexample does not satisfy the target (there were no astronauts born in Antarctica) because it satisfies our hypothesis. The Horn Envelope algorithm tests all intersections of received negative counterexamples using membership queries. In this case, the intersection of $\mathcal{I}$ and $\mathcal{J}$ is $\mathcal{I} \cap \mathcal{J} = \{\mathsf{astronaut}\}$. The algorithm would then ask a

---

[1]The inclusion of "Do not specify which values are different" in the prompt was motivated by the models tendency to explain its answer in unpredictable ways. These explanations made parsing the response more difficult.

membership query with this intersection $\mathsf{MQ}_\mathcal{T}(\{\mathsf{astronaut}\})$. Suppose that the answer is 'Yes'. Then the algorithm considers the interpretation $\{\mathsf{astronaut}\}$ as one that satisfies the target. Then, the rule $\neg(\mathsf{astronaut})$ is *not* added to the hypothesis. So, the counterexample $\mathcal{J} = \{\mathsf{astronaut}, \mathsf{Antartica}\}$ is unchanged and the algorithm adds the rule $\neg(\mathsf{astronaut} \wedge \mathsf{Antarctica})$ to its hypothesis. The updated hypothesis would be: $\{\neg(\mathsf{astronaut} \wedge \mathsf{Antarctica}) \wedge \neg(\mathsf{astronaut} \wedge \mathsf{before\_1900})\}$

Testing intersections helps to create more informative counterexamples. When the model is no longer able to give a counterexample, the hypothesis is considered to be equivalent to the Horn envelope of the target, which is the closest Horn approximation of the propositional expression representing the relationship between the variables in the model. We note that rules of the form $(\mathsf{astronaut} \wedge \mathsf{before\_1930}) \rightarrow \mathsf{male}$ (expressing that 'astronauts born before 1930 are male') can also be generated by the algorithm. This can happen when an interpretation that is not satisfied by the target is contained in an interpretation that is satisfied.

**Validating EQ responses** While LLMs are capable of responding to queries they do not always respond with a valid counterexample following the requested format. To ensure the validity of the responses to the equivalence queries, we check and handle a number of ways the responses might be invalid. The full list of steps can be seen in Algorithm 1, and a taxonomy of the validation steps can be seen in Table 1. If the validation checks discovers an error, then the EQ prompt is modified to reflect the error and the LLMs is queried again for a counterexample. In the case where the LLM is unable to understand the error description, to avoid repetitive loops, a threshold is set such that if the LLM makes too many validation errors in a row[2], then we consider the model unable to provide a valid counterexample and terminate the run.

---

**Algorithm 1:** Equivalence Query Response Validation

---

**input** : Equivalence Oracle $\mathsf{EQ}_\mathcal{T}^{\mathsf{Horn}}(\mathcal{H})$, hypothesis $\mathcal{H}$, list of previous counterexamples $L$
**output** : Validated *counterexample* or hypothesis $\mathcal{H}$

1   prompt $\leftarrow$ translate $\mathcal{H}$ to natural language and ask for counterexamples
2   **while** *the error threshold is not reached* **do**
3     response $\leftarrow$ query $\mathsf{EQ}_\mathcal{T}^{\mathsf{Horn}}(\mathcal{H})$ with prompt
4     **if** *done thinking* **then**
5       **if** *response is to terminate* **then**
6         **return** ($\mathcal{H}$, *termination flag*)
7       **if** *response can be parsed* **then**
8         potential counterexample $\leftarrow$ parse response
9         **if** *potential counterexample is not a duplicate* **then**
10           **if** *potential counterexample is logically valid* **then**
11             **return** validated counterexample

12    append response with description of error to the prompt
13 **return** ($\mathcal{H}$, *non-termination flag*)

---

## 5   EXPERIMENTS: GENDER-OCCUPATION RELATIONSHIPS IN LLMS

All experiments were performed on a High-Performance Computing (HPC) cluster with 1 GPU-accelerated node per experiment. The specific GPUs were either a NVIDIA A100 or a NVIDIA RTX3090, limited to 90GB of memory per run.

To allow a meaningful comparison with earlier research, we consider the same occupational gender bias scenario presented in Blum et al. (2024). The authors generate sentences using a template with a

---

[2]In our experiments the threshold of consecutive errors was set to 10.

Table 1: Taxonomy of response validation

| Validation step | Definition |
|---|---|
| *done thinking (4)* | The Deepseek reasoning model used in the experiments prepares its responses with a set of <think> tags. If there is no closing tag then the model did not complete its thinking, not providing a valid response. |
| *terminate (5)* | If the termination phrase is in the response then we return the current hypothesis $\mathcal{H}$. |
| *parse (7)* | If the response follows the requested response format, only contains values from the allowed set, and the response does not contain multiple counterexamples, then we parse the potential counterexample from natural language to logical language. |
| *duplicate (9)* | If the potential counterexample is not a duplicate of a previously given counterexample. |
| *valid (10)* | If the counterexample is a valid counterexample, that is, if the counterexample is in the symmetric difference between the hypothesis and target. |

limited set of values for each variable[3] seen in Table 2. We recreate their experiments[4] using their method, referred to in this paper as the *sampling method*, and compare with our own *direct EQ* approach.

Table 2: Set of allowed values

| Variable | Values |
|---|---|
| **Continent** | 'Africa', 'Americas', 'Asia', 'Europe', 'North America', 'Oceania', 'South America', 'Australia', or 'Eurasia'. |
| **Time Period** | 'before 1875', 'between 1875 and 1925', 'between 1925 and 1951', 'between 1951 and 1970', or 'after 1970' |
| **Occupation** | 'fashion designer', 'nurse', 'dancer', 'priest', 'footballer', 'banker', 'singer', 'lawyer', 'mathematician', or 'diplomat' |
| **Gender** | 'woman', or 'man' |

**Direct EQ** In the direct EQ approach we generate a prompt that communicates our current hypothesis and elicits counterexamples from the model directly. To avoid influencing the model by giving examples of possible and/or impossible combinations, we employed a zero-shot strategy for the first counterexample communicating the zero-hypothesis (everything is possible), instructions describing the format of the replies, and restrictions on the domain found in Table 2. We ask the model to provide us with a counterexample in the following form: "`A person born in <continent>, <time period>, who is a <occupation> CAN/CANNOT be a <gender>.`" Once a counterexample has been given, both the updated hypothesis and all previous counterexamples are communicated to the model in a few-shot setting.

**Models** For the experiments with the sampling method we used the RoBERTa-base and RoBERTa-large models (Liu et al., 2019) also used by Blum et al. (2024) as well as the instruction tuned Mistral-7B (Jiang et al., 2023) generative model. For the experiments with the direct EQ approach we used the instruction tuned Mistral-7B and Mistral-24B modelsas well as the DeepseekR1-8B reasoning model distilled from Llama (DeepSeek-AI, 2025).

## 6 EVALUATION AND RESULTS

We evaluate the resulting hypotheses by applying the following three criteria.

**Number of Queries** Prompting generative LLMs can be resource intensive. The sampling approach used by Blum et al. (2024) may generate many examples before a valid counterexample is found. A

---

[3]They also include 'unknown value', relevant to the Horn algorithm but unnecessary for our work as counterexamples generated with our approach simply omits variables that are not set. E.g. "A person born `time period` in `continent`." if the gender is not set.

[4]We made the change to run their experiments to termination instead of limiting the experiment to 200 equivalence queries as was done by Blum et al. (2024).

formula for the upper bound of the number of queries needed is given by the Probably Approximately Correct (PAC) framework(Valiant, 1984; Angluin, 1987). The formula is $\frac{ln(\frac{|H|}{\delta})}{\epsilon}$ where $|H|$ is the size of the hypothesis space (that is, the set of all possible hypotheses), $\epsilon$ represents the error, and $\delta$ represents the confidence.[5] This upper bound can be quite large, so reducing the number of queries needed to generate the hypothesis is an important aspect of the evaluation.

**Consistency of Hypotheses**   Consistency of the hypotheses across multiple runs helps build confidence that the extracted hypothesis is equivalent to the target. To determine whether the model-generated hypotheses really represent the target, we evaluate the variance in the similarity of the extracted hypotheses. We evaluate the similarity by directly comparing clauses of the hypotheses for equality. Logically equivalent rules are treated as equal (e.g. $\neg(p \wedge q)$ and $\neg(q \wedge p)$ are logically equivalent and treated as equal in our calculation). Larger intersections between pairs of hypotheses with smaller variance indicate more consistent models.

As our baseline, we consider the expected size of the intersection of extracted rules between each pair of runs. Let $S_1, S_2$ denote their sets of rules, which are subsets of the set $R$ of all possible rules.Assume that $S_1, S_2$ are random subsets of $R$. The expected size of the intersection $S_1 \cap S_2$ given the sizes of $S_1$ and $S_2$ is

$$E(|S_1 \cap S_2| \mid |S_1|, |S_2|) = \frac{|S_1| \cdot |S_2|}{|R|} \tag{1}$$

We compare the expected size of the intersection with the actual size of the intersection to determine whether the intersection could be attributed to randomness.

**Correlation of rules with historical data**   Wikidata[6] contains publicly available data about historical persons including their occupations. We compare how the set of rules extracted from the LLMs corresponds with the data extracted from Wikidata. We should be aware that Wikidata contains historical data, which in some cases reflect historical gender biases with respect to occupation. Some examples of such biases can be seen in Figure 5 in the appendix.

To study the correspondence between the rules extracted and real world information present in large datasets such as Wikidata, we calculate the *confidence* of the rule commonly used in association rule learning (Agrawal et al., 1993). The confidence is calculated with the following equation.

$$\text{conf}(X \rightarrow Y) = \frac{\text{supp}(X \cap Y)}{\text{supp}(X)} \tag{2}$$

The confidence of a rule is calculated by counting the number of people in Wikidata that support both the antecedent and the consequent of the rule, divided by the number of people that support the antecedent[7]. Since the gender in our experiments is a binary variable limited to male or female, we consider that rules of the form $\neg(\text{continent} \wedge \text{time\_period} \wedge \text{occupation} \wedge \text{woman})$ equivalent to $(\text{continent} \wedge \text{time\_period} \wedge \text{occupation}) \rightarrow \text{man}$ and vice-versa.

In the following paragraphs we provide an analysis of the results of our experiments.

**Number of queries** One of the biggest limitations of previous methods is that the number of queries required before a counterexample is found grows exponentially as the hypothesis approximates the target, as seen in Figure 1. In contrast, the longest running direct EQ run was with the Mistral(24B) model which required an average of 46.7 queries before termination as seen in Table 3. This is a dramatic reduction in the number of queries needed.

**Consistency of Hypotheses** The standard deviations on the number of queries and in the size of the intersections show that the direct EQ method has more variability across runs. For the smaller Mistral(7B) model, terminating after consecutive errors inflates the query count as the model is

---

[5]A bug in the code was found where the $log_2$ was used instead of the natural log. This results in a slightly lower threshold in the experiments.

[6]www.wikidata.org

[7]During our calculations of the confidence of extracted rules, the Mistral(7B) model using the direct EQ approach extracted the rule $\neg(\text{before\_1875})$ which we removed before calculating the confidence.

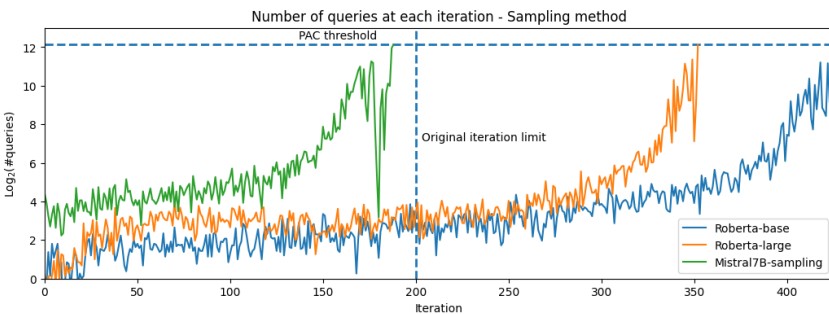

Figure 1: $\text{Log}_2$ of the number of queries per iteration using the sampling method. The number of queries needed grows exponentially as the size of the hypothesis grows. The threshold for termination was set using PAC framework

Table 3: Size of Horn($\mathcal{H}$) and non-Horn($\mathcal{Q}$) rules, intersections, expected intersections, average weighted confidence, and the number of queries for both the sampling and the direct EQ methods.

| Sampling method | | | | | | |
|---|---|---|---|---|---|---|
| Model | Size($\mathcal{H}$) | $|\cap|$ | E($|\cap|$) | Conf | #Queries | Size($\mathcal{Q}$) |
| RoBERTa-B | 94.00±0.00 | 94.00±0.00 | 9.82±0.00 | 0.48±0.32 | 14,839±1921 | 91.0±0.0 |
| RoBERTa-L | 110.90±0.30 | 110.80±0.40 | 13.67±0.05 | 0.74±0.25 | 13,140±1470 | 63.9±0.3 |
| Mistral(7B) | 55.62±0.78 | 53.85±1.86 | 1.56±0.03 | 0.58±0.46 | 18,271±2245 | 28.5±1.2 |
| Direct EQ method | | | | | | |
| Model | Size($\mathcal{H}$) | $|\cap|$ | E($|\cap|$) | Conf | #Queries | Size($\mathcal{Q}$) |
| Mistral(7B) | 9.70±2.69 | 2.53±1.33 | 0.10±0.04 | 0.50±0.32 | 11.5±10.9 | 0.1±0.4 |
| Mistral(24B) | 15.75±6.66 | 3.42±1.76 | 0.27±0.16 | 0.84±0.34 | 46.7±23.0 | 0.0±0.0 |
| DeepSeek | 3.10±1.67 | 0.15±0.43 | 0.00±0.00 | 0.84±0.34 | 7.8± 9.8 | 0.1±0.2 |

often not able to parse the error description well and the model repeats the same mistake again. This repetitive behaviour was what motivated us to include this error threshold. This premature termination behaviour results in that the size of the extracted hypotheses for the smaller Mistral(7B) model is much smaller when using the direct EQ method. The larger Mistral(24B) model and the Deepseek reasoning model were better able to follow instructions and therefore more likely to terminate intentionally. The Deepseek model often searched the hypothesis space within its <think> tags, rejecting many possible counterexamples in the process. One text extracted from the think tag of a run with the Deepseek model has been provided for illustration purposes and can be found in appendix A.5. This highly selective behaviour likely influenced the small size of the resulting hypotheses and the low consistency for the reasoning model. The size of the intersections across runs are much larger than the expected intersections for both approaches showing that both approaches are able to extract common rules at a rate well above random chance.

**Correlation and confidence** In Table 4 we see a subset of the rules with the highest confidence extracted with the sampling method, and Table 5 shows the same for models with the direct EQ approach. Both approaches generate very confident rules but the sampling approach also has a tendency to generate many low confidence rules, seen in Tables 6, 7, and 8 in the appendix. This and the sizes of the extracted hypotheses indicates that the sampling approach are more likely to accept an example as a counterexample than the direct EQ approach. This is supported by the size of the non-Horn rules $\mathcal{Q}$ which is generated along with the hypothesis $\mathcal{H}$ by Algorithm 2 for each run. For a rule to appear in $\mathcal{Q}$, it first needs to be appear in $\mathcal{H}$, before being proven to be representing a non-Horn clause by later positive counterexamples. From the average size of $\mathcal{Q}$ for the sampling method we see that many rules extracted were later removed from $\mathcal{H}$ by the algorithm. The remaining low confidence rules may be non-Horn rules that were not removed from $\mathcal{H}$ before hitting the PAC threshold. These low confidence rules bring down the weighted average confidence of the extracted rules for the sampling approach.

Table 4: Five most confident rules using the sampling method.

| Model | Rules | Rate | Conf |
|---|---|---|---|
| RoBERTa-base | ¬(Australia ∧ before 1875 ∧ lawyer ∧ woman) | 1.00 | 1.00 |
| | ¬(priest ∧ woman) | 1.00 | 0.98 |
| | ¬(before 1875 ∧ mathematician ∧ woman) | 1.00 | 0.98 |
| | ¬(between 1875 and 1925 ∧ diplomat ∧ woman) | 1.00 | 0.98 |
| | ¬(banker ∧ woman) | 1.00 | 0.96 |
| RoBERTa-large | ¬(before 1875 ∧ diplomat ∧ woman) | 1.00 | 1.00 |
| | ¬(before 1875 ∧ lawyer ∧ woman) | 1.00 | 0.99 |
| | ¬(S. America ∧ before 1875 ∧ woman) | 1.00 | 0.99 |
| | ¬(priest ∧ woman) | 1.00 | 0.98 |
| | ¬(before 1875 ∧ mathematician ∧ woman) | 1.00 | 0.98 |
| Mistral(7B) | ¬(Oceania ∧ before 1875 ∧ mathematician ∧ woman) | 1.00 | 1.00 |
| | ¬(Australia ∧ before 1875 ∧ man ∧ nurse) | 1.00 | 1.00 |
| | (Asia ∧ before 1875 ∧ footballer) → man | 1.00 | 1.00 |
| | (Australia ∧ before 1875 ∧ diplomat) → man | 1.00 | 1.00 |
| | ¬(Africa ∧ between 1875 and 1925 ∧ footballer ∧ woman) | 1.00 | 1.00 |

Table 5: Five most confident rules using the direct EQ method.

| Model | Rule | Rate | Conf |
|---|---|---|---|
| Mistral(7B) | ¬(N. America ∧ priest ∧ woman) | 0.05 | 0.96 |
| | ¬(Oceania ∧ man ∧ nurse) | 0.05 | 0.96 |
| | ¬(S. America ∧ between 1925 and 1951 ∧ mathematician ∧ woman) | 0.05 | 0.87 |
| | ¬(Africa ∧ diplomat ∧ woman) | 0.05 | 0.85 |
| | ¬(Europe ∧ after 1970 ∧ dancer ∧ man) | 0.05 | 0.59 |
| Mistral(24B) | ¬(Africa ∧ before 1875 ∧ man ∧ nurse) | 0.85 | 1.00 |
| | ¬(S. America ∧ banker ∧ before 1875 ∧ woman) | 0.25 | 1.00 |
| | ¬(Africa ∧ banker ∧ before 1875 ∧ woman) | 0.20 | 1.00 |
| | ¬(S. America ∧ before 1875 ∧ mathematician ∧ woman) | 0.15 | 1.00 |
| | ¬(Australia ∧ before 1875 ∧ footballer ∧ woman) | 0.10 | 1.00 |
| Deepseek(8B) | ¬(Europe ∧ before 1875 ∧ priest ∧ woman) | 0.11 | 1.00 |
| | ¬(Americas ∧ between 1875 and 1925 ∧ footballer ∧ woman) | 0.05 | 1.00 |
| | ¬(Eurasia ∧ before 1875 ∧ lawyer ∧ woman) | 0.05 | 1.00 |
| | ¬(Europe ∧ before 1875 ∧ lawyer ∧ woman) | 0.11 | 1.00 |
| | ¬(Europe ∧ between 1875 and 1925 ∧ priest ∧ woman) | 0.11 | 0.99 |

## 7 CONCLUSIONS

In this paper we explore a new strategy for extracting Horn rules from large language models. We propose an evaluation scheme for the extracted Horn rules using three criteria and compare results with previously proposed strategies used on the BERT family of models. We find that our new strategy significantly reduces the number of queries required while extracting on average higher confidence rules. The significant number of queries required with the sampling method makes it infeasible for use with generative LLMs. Our new strategy thus provides an alternative method for extracting Horn rules from generative LLMs.

**Limitations and Future Work** A limitation of this work is the scope of the experiments; the number and type of models tested. While the case study was meant to showcase an application of the strategy, we encountered challenges with prompt sensitivity, leading to diverging results, as well as challenges interpreting responses as smaller models were not able to process the instructions in the prompt and provide valid counterexamples as the set of extracted rules grew larger. Another limitation of this work is that we do not address how the non-deterministic behaviour of generative models impact the guarantees of Algorithm 2. As future work, we would like to investigate other case studies since the approach is independent of the variables in Table 2, improve the expressivity of the rule language, and provide a formal treatment for the changes in the answers of the LLMs.

## 7.1 ETHICS STATEMENT

This work extracts rules on gender-occupation relationships in LLMs. We compare the rules extracted against historical data which in some cases reflect historical gender biases. We wish to acknowledge that when working with such biases it is important to emphasize that it is not our intention to perpetuate historical biases. With this work, we only wish to contribute to the removal of harmful biases by highlighting their existence.

## 7.2 REPRODUCIBILITY STATEMENT

The code used to run the experiments is added as supplementary material. A description of each file in the repository is included in the README.md. All experiments were performed on a High-Performance Computing (HPC) cluster with 1 GPU-accelerated node per experiment. The specific GPUs were either a NVIDIA A100 or a NVIDIA RTX3090, limited to 90GB of memory per run. The historical data used to compare the results were extracted from, and is available on `www.wikidata.org`. The code for the extraction of the data is available in the code in the supplementary material.

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
