# OpenReview forum: "Actively Learning Horn Envelopes from LLMs"
_ICLR.cc/2026/Conference — ICLR 2026 Conference Withdrawn Submission_

### Official Review · Reviewer_YFMk · 2025-10-25

**Soundness:** 3
**Presentation:** 2
**Contribution:** 2
**Rating:** 2
**Confidence:** 3

**Summary:**

The paper explores rule extraction from large language models within the framework of exact concept learning. Prior work simulated equivalence queries through random sampling, which is inefficient and often redundant. The authors propose to instead directly prompt the model to produce counterexamples, effectively operationalizing equivalence queries rather than approximating them. They combine this with a validator step that filters malformed or inconsistent outputs before integrating them into the hypothesis. The study evaluates this approach, that the authors call Direct EQ, on a gender–occupation dataset used by prior work, comparing it to sampling-based methods. Results show orders-of-magnitude fewer queries, smaller learned rule sets, and higher cross-run consistency over the sampling-based method on relatively current Mistral and DeepSeek models.

**Strengths:**

- The proposal is practical and implementable. The DirectEQ approach,prompting for counterexamples, is straightforward and fairly model-agnostic (as much as prompting strategies are). The submission thus presents a practical applied idea for a timely problem.

- There are clear and natural benefits over prior sampling based work in terms of query efficiency. Given the notable ressource usage of LLMs I consider this to be a very important axis of improvement.

**Weaknesses:**

- The reporting of experimental results is somewhat opaque. For example, in Table 3 it is unclear what the central takeaway is meant to be. While the number of required queries is clearly lower, the effect on the other reported metrics is hard to interpret. With so many indicators and ratios, it is not obvious to a reader outside this sub-area what constitutes a good outcome and what the tradeoff between number of queries and those outcomes is.

-  The proposed DirectEQ approach is evaluated on different models than the prior sampling approach. This can skew the results, and in fact on the one overlapping model (Mistral 7b) the sampling method seems to generate more hypothesis at higher confidence? It is entirely unclear to me whether the improvements in any metric except #queries in Table 3 comes from using stronger LLMs or whether is a result of the Direct EQ method.

- The conceptual contribution feels limited. In slightly simplified terms, instead of sampling for counter-examples, the submission suggests a prompting regime to ask whether there are counterexamples. While this seems naturally more efficient, it also seems like a rather small conceptual step and I struggle to see any major contribution that elevates the submission to the level of ICLR.

- The method description puts emphasis on the importance of the validation step and promises details in Algorithm 1. There however is only a chain of ifs checking whether the response can be parsed, whether it is a duplicate, and whether it is "valid" with no further details. Table 1 also provides no actionable insight to me (the footnotes in the table also all seem broken/lost). This level of detail is insufficient for reproduction and weakens the claim of a significant technical contribution.

**Questions:**

Can you elaborate on why you recall the PAC learning generalisation error bound here? It is not clear to me how the bound applies to this setting as it assumes a perfect oracle, fixed hypothesis size, and i.i.d. random samples. It is unclear to me how the setting satisfies and of these three points.

Are there any other approaches to rule learning from LLMs? I recall a few conversations with colleagues on the topic over the last few years and I am therefore surprised that the sampling based approach of Blum et al. is the only prominently features alternative in the literature so far?

---

### Official Review · Reviewer_ZZ5U · 2025-10-30

**Soundness:** 2
**Presentation:** 3
**Contribution:** 1
**Rating:** 0
**Confidence:** 4

**Summary:**

This paper proposes a strategy for extracting Horn rules from Large Language Models (LLMs) using Angluin’s exact learning framework. The authors adapt a previously proposed method to simulate equivalence queries, leveraging LLMs’ generative capabilities to elicit counterexamples. While the goal of improving explainability in black-box LLMs is relevant and commendable, the paper’s contributions feel incremental rather than novel. The approach is essentially a minor extension of prior work by Blum et al. [1], applied to a few additional language models without substantial innovation.
The experimental evaluation is highly limited, focusing on a single, relatively simple case study (gender-occupation relationships) and only a couple of small LLMs. This raises concerns about the method’s generalizability and robustness, particularly given the lack of analysis on how LLM hallucinations might affect the quality of extracted rules. Additionally, the paper overlooks the critical role of prompt engineering, which could significantly impact results. While the availability of source code is a positive aspect, the overall contribution feels more like an engineering effort than a meaningful research advancement, especially for a conference like ICLR.

[1]. Sophie Blum, Raoul Koudijs, Ana Ozaki, and Samia Touileb. Learning horn envelopes via queries from language models. Int. J. Approx. Reason., 171:109026, 2024. doi: 10.1016/J.IJAR.2023. 109026. URL https://doi.org/10.1016/j.ijar.2023.109026.

**Strengths:**

- The paper addresses a critical issue in LLM research: explainability. Extracting Horn rules from LLMs is a valuable step toward interpreting their decision-making processes.
- The source code is available, which enhances the reproducibility of the results and allows the community to build on this work.

**Weaknesses:**

- The proposed approach is a direct and minor extension of the work by Blum et al. [1]. The authors do not introduce significant innovations, making the contribution feel incremental rather than groundbreaking.
- The paper primarily focuses on prompt engineering and output formatting to extract Horn clauses. While these are necessary steps, they do not constitute a substantial research contribution, particularly for a high-impact conference like ICLR.
- The evaluation is restricted to a single, simple case study (gender-occupation relationships) and only a few small language models. This raises concerns about the method’s applicability to more complex tasks or larger models. The authors should extend their experiments to include a broader range of LLMs and more challenging tasks to demonstrate the method’s robustness and scalability.
- The approach does not account for LLM hallucinations, which could lead to logically valid but incorrect counterexamples. This issue could significantly degrade the quality of the extracted Horn rules and should be thoroughly analyzed and discussed. The authors should manually verify the frequency of hallucinations in their experiments and discuss their impact on the results.
- The paper does not address the critical role of prompt engineering in the results. Different prompts could lead to drastically different outcomes, yet the authors do not provide insights into how prompts were constructed, optimized, or tested for generalizability.

**Questions:**

- The approach seems to be a minor extension of Blum et al. [1]. Could the authors clarify what specific improvements or innovations their method introduces beyond the prior work?
- Given that the contribution feels more like an engineering effort, how do the authors envision this work advancing the field of explainable AI or LLM research in a meaningful way?
- Why was the evaluation limited to a single, relatively simple case study? Could the authors discuss how the method might perform on more complex tasks or domains?
- The experiments only include a couple of small LLMs. How might the method scale to larger or more diverse models, and what challenges might arise?
- How do the authors plan to address the issue of LLM hallucinations generating incorrect counterexamples? Could they provide an analysis of how often this occurs in their experiments and its impact on rule quality?
- The paper does not discuss the impact of prompt engineering on the results. Could the authors provide insights into how prompts were designed, optimized, and tested for robustness?
- How generalizable is the approach to different prompts or prompt formulations? Could the authors share results or analyses demonstrating consistency across varied prompts?
- The paper lacks a formal analysis of how non-deterministic LLM responses affect the correctness of the algorithm. Could the authors discuss potential ways to formalize or mitigate this issue?
- How does this method compare to other explainability techniques (e.g., attention analysis) in terms of efficiency, accuracy, and applicability?
- Could this approach be extended to other logical formalisms or more complex rule types? What challenges might arise in doing so?

---

### Official Review · Reviewer_DnJh · 2025-10-30

**Soundness:** 2
**Presentation:** 2
**Contribution:** 2
**Rating:** 2
**Confidence:** 3

**Summary:**

The paper presents an approach to extract structured knowledge by querying LLMs. In particular, the approach learns a horn envelope to approximate the target by querying the LLMs with two types of queries, membership queries which yield binary answers and equivalence queries which yield counterexamples. The membership queries uncover the knowledge encoded within the LLM to refine the hypotheses and the equivalence queries test if the current hypotheses match the target and if not use counterexamples from the LLM to incrementally move closer to the target. Experiments are performed on a gender bias scenario proposed in an earlier work and the rules generated are compared to those generated through sampling BERT embeddings in an earlier work.

**Strengths:**

+ Using LLMs to generate structured knowledge bases seems interesting and should have several applications
+ The approach successfully limits the number of queries to an LLM (which can be expensive) to learn a non-trivial target concept
+ The formalism of connecting LLMs to learn a Horn envelope using Angluin’s learning framework seems novel

**Weaknesses:**

- The main weakness is the limited evaluation which shows the results specific to a single target scenario. The method seems general enough to be applicable, therefore, a more convincing empirical evaluation would strengthen the paper
- The comparison against a single approach, i.e., the random sampling method seems a bit limited to show significance. Particularly since when using LLMs we can directly generate counterexamples. A stronger baseline is perhaps other approaches that use LLMs for rule learning(e.g. Large Language Models can Learn Rules Zhu et al.).
- Since LLMs are inherently stochastic, would the rules learned be very different in each run. Lines 418-419 says “better than random chance” I was not sure if this means the same rules are learned with probability > 50%. Also, depending on the temperature settings, the LLM can give very different answers, should this be taken into account?

In general, the ideas seem to be good but may need further exploration and stronger empirical validation.

**Questions:**

Are there other rule learners using LLMs? If so, how does the proposed approach compare with them both conceptually and empirically.

---

### Official Review · Reviewer_qtKq · 2025-11-01

**Soundness:** 2
**Presentation:** 2
**Contribution:** 2
**Rating:** 2
**Confidence:** 3

**Summary:**

This paper proposes a framework for actively extracting Horn rules (logical implications) from large language models (LLMs) using a modification of Angluin’s exact learning framework. For implementing equivalence queries, this paper proposes a direct, adaptive prompting strategy that asks the LLM to generate a counterexample when a hypothesis is incorrect.

**Strengths:**

1. Introduces direct equivalence querying using LLM generation to obtain counterexamples.
2. Implements an elaborate response validation pipeline to handle malformed or inconsistent replies.
3. Demonstrates substantial efficiency gains (two orders of magnitude fewer queries) while maintaining rule quality.
4. Comparing extracted rules to Wikidata illustrates alignment with human-verified knowledge.

**Weaknesses:**

1. The EQ mechanism relies on prompting and is therefore fragile, especially for the relatively smaller models that fail to parse hypotheses or generate valid counterexamples. As a result, there is no formal correctness guarantees (the Angluin’s theoretical assumptions that the teacher is guaranteed to produce minimal counterexamples), so convergence to the true Horn envelope isn’t formally proven. In addition, LLM’s counterexamples are not verifiably in H \oplus T because we do not have ground truth of LLM's world view. The paper treats the LLM’s response as ground truth, but it lacks an oracle guarantee, logical entailment checking, or independent verification step.
2. Limited generalization and domain scope. The work only conducts one case study (gender–occupation) and has no demonstration that the approach scales to more complex relations or domains (e.g., multi-variable reasoning, temporal or causal rules).
3. The validation overhead needs to be discussed. Re-prompting for malformed outputs introduces hidden query costs.
4. The evaluation on gender-occupation rules may be confounded by model's internal gender bias.
5. The dataset only consists of 10 occupations, 2 genders, 5 time periods. Such a small symbolic space limits insight into scalability.

**Questions:**

Algorithm 1 stops either when no counterexample can actually be found or when the LLM fails 10 times. How can we distinguish true epistemic convergence from practical timeout?

---

### Note · Authors · 2025-11-14

I have read and agree with the venue's withdrawal policy on behalf of myself and my co-authors.